# Analysis of the Anti-Inflammatory Capacity of Bone Broth in a Murine Model of Ulcerative Colitis

**DOI:** 10.3390/medicina57111138

**Published:** 2021-10-20

**Authors:** Laura M. Mar-Solís, Adolfo Soto-Domínguez, Luis E. Rodríguez-Tovar, Humberto Rodríguez-Rocha, Aracely García-García, Víctor E. Aguirre-Arzola, Diana E. Zamora-Ávila, Aime J. Garza-Arredondo, Uziel Castillo-Velázquez

**Affiliations:** 1Laboratory of Veterinary Immunology, School of Veterinary Medicine and Zootechnics, Autonomous University of Nuevo León, General Escobedo 66050, Nuevo León, Mexico; lauramarisolmar@gmail.com (L.M.M.-S.); ledgart@hotmail.com (L.E.R.-T.); diana.zamoravl@uanl.edu.mx (D.E.Z.-Á.); aime_jazmin@hotmail.com (A.J.G.-A.); 2Department of Histology, School of Medicine, Autonomous University of Nuevo León, Monterrey 64460, Nuevo León, Mexico; adolfo.sotodmn@uanl.edu.mx (A.S.-D.); humbertordz54@gmail.com (H.R.-R.); aracely_79_20@yahoo.com (A.G.-G.); 3Laboratory of Chemistry and Biochemistry, School of Agronomy, Autonomous University of Nuevo León, General Escobedo 66050, Nuevo León, Mexico; veaguirre@gmail.com

**Keywords:** bone broth, nutritional composition, essential amino acids, ulcerative colitis, cytokines

## Abstract

*Background and Objectives:* Nutritional deficiencies are one of the main triggers for the development of gastrointestinal diseases, such as ulcerative colitis (UC). Therefore, the objective of the present work consisted of determining the nutrients present in the bone broth (BB) and evaluating their anti-inflammatory properties in a murine model of UC, induced by intrarectal administration of 2, 4, 6-trinitrobenzene sulfonic acid (TNBS), and acetic acid (AcOH). The BB was prepared from the femur of bovine cattle and cooked in distilled water for 8 h at 100 ± 2 °C. *Materials and Methods:* The BB was administered ad libitum to BALB/c mice for 10 days before the induction of UC. Colon samples were collected for histological analysis and determination of cytokine expression levels by qPCR. *Results:* It was found that amino acids (AA) are the main nutritional contribution of BB, 54.56% of these correspond to essential AA. The prophylactic administration of BB in the murine model of UC reduced histological damage, decreased the expression of *IL-1β* (61.12%), *IL-6* (94.70%), and *TNF-α* (68.88%), and increased the expression of *INF-γ* (177.06%), *IL-4* (541.36%), and *IL-10* (531.97%). *Conclusions:* This study shows that BB has anti-inflammatory properties, and its consumption can decrease the symptoms of UC.

## 1. Introduction

In recent years, it has been observed that nutritional deficiencies in the diet of industrialized countries are the main risk factor for the development of various diseases, that affect the gastrointestinal tract [1,2], such as ulcerative colitis (UC) [3,4,5].

The UC is a chronic, idiopathic and recurrent disease that causes inflammation and morphological changes in the mucosa and submucosa of the colon and rectum [6,7]. On the other hand, it is believed that the onset, severity, and perpetuation of the disease is due to a dysregulation of the immune response [8]. During UC the expression of pro-inflammatory cytokines is increased, these are the cause of the lesions in the mucosa and tissue damage in the colon, the pro-inflammatory cytokines identified in UC are *TNF-α*, *IL-1-β, IL-6,* and *INF-γ* [9]. Although the organism has mechanisms that help counteract inflammation, such as the secretion of anti-inflammatory cytokines: *IL-4* and *IL-10*, their expression is almost nil during periods of disease activity [10]. At the histological level, UC causes infiltration of lymphocytes, plasma cells, and granulocytes, loss of goblet cells, ulcerations, and distortion of the crypts of the colon and rectum mucosa [11].

Currently, the drugs used for the treatment of this disease are based on the administration of anti-inflammatory and immunosuppressive drugs for long periods, the disadvantage of these is that they are expensive, and have a low degree of effectiveness, with several side effects [12,13], unlike therapies based on a nutritional approach have proven to be effective for the treatment and prevention of chronic diseases, without causing side effects [5]. Because food supplements rich in amino acids (AA) and minerals are expensive, they are not a profitable alternative, so a low-cost alternative to combat this disease is the consumption of foods rich in nutrients, such as by-products of animal origin [14].

Among the foods of animal origin that provide a large amount of nutrients is bone broth (BB). It has been reported that by long-lasting cooking of bones it is possible to obtain a large amount of AA, minerals, and proteins such as collagen [15,16,17,18,19]. The therapeutic properties of this food have not been fully studied by scientific methods, but in countries such as Mongolia, it is consumed to stimulate the immune system and promote the proper functioning of the digestive system, these therapeutic properties being attributed to its nutritional content [1,15,17].

Currently, there are no studies that evaluated the therapeutic effect of BB in UC. Therefore, the objective of the present work consisted of determining the nutrients present in the BB, and evaluating their anti-inflammatory properties in a murine model of UC, at histological and molecular levels.

## 2. Materials and Methods

### 2.1. Preparation of the Bone Broth

BB was made from bovine femur, the bones were obtained commercially and crosswise cut to obtain pieces of 100–130 g.

To eliminate the remains of meat, fat, and blood present in the bones, the pieces were left to stand in distilled water at 50 °C for 15 min, the water was discarded at the end of the wash [17]. This procedure was performed in triplicate.

The preparation of the BB was performed using a slow cooker (Taurus, Oliana, Spain). The preparation was performed in a ratio of 1:4 concerning the weight of the bones used. We use acidified water that was made by mixing 20 mL of white vinegar with 1 L of distilled water, this water was used to simmer the broths. The water was brought to the boiling point (100 °C) before the incorporation of the bones. The cooking was maintained for 8 h at 100 ± 2 °C. Acidified distilled water was added to maintain the initial volume of the preparation [16].

At the end of the cooking process, the preparation was left to cool at room temperature for 3 h, and then it was refrigerated at 4 °C for 6 h. After this time, the BB was filtered to remove the bones and fat resulting from the cooking process using a funnel with a cotton gauze. Finally, the BB was preserved at −20 °C for its subsequent use and analysis.

### 2.2. Determination of the Nutritional Content of the Bone Broth

Before the determination of protein and AA content, the BB was subjected to a lyophilization process. For the analysis of minerals, a liquid sample of BB was used. All analyses were performed in triplicate.

#### 2.2.1. Protein

The amount of total nitrogen present in 1 g of BB was determined by the Dumas method employing an elemental analyzer by combustion TRUSPEC CHN (Leco Corp., St. Joseph, MI, USA). The amount of protein present in the sample was calculated by multiplying the amount of total nitrogen obtained by the conversion factor 6.25 [20].

#### 2.2.2. Amino Acids

The contents of aspartic acid (Asp), glutamic acid (Glu), serine (Ser), glycine (Gly), lysine (Lys), histidine (His), threonine (Thr), arginine (Arg), alanine (Ala), proline (Pro), cysteine (Cys), tyrosine (Tyr), valine (Val), methionine (Met), isoleucine (Ile), leucine (Leu) y phenylalanine (Phe) were determined from 1 g of BB by high-performance liquid chromatography (HPLC), using the AccQ-Tag Kit (Waters Chromatography, S.A., Barcelona, España, following the manufacturer′s instructions.

#### 2.2.3. Minerals

The contents of calcium (Ca), magnesium (Mg), phosphorus (P), sodium (Na), potassium (K), iron (Fe), copper (Cu), zinc (Zn), manganese (Mn), cobalt (Co), and molybdenum (Mo) were determined from 100 mL of BB by inductively coupled plasma optical emission spectrophotometry. The analysis was performed according to the official method 2011.14 of the Association of Official Analytical Chemists (AOAC International, Rockville, MD, USA, 2016).

##### Determination of Hazard Ratio

To evaluate the benefits or risks associated with the ingestion of metals present in BB, we determined the hazard ratio (HR), which consisted of determining the amount of mg of the mineral of interest consumed in every 100 mL of BB (maximum daily intake volume), the mg obtained was divided by the amount of recommended mg of daily intake of the mineral of interest, a value of HR > 1 indicates an increase in the probability that the ingested dose of that mineral causes a health risk. On the other hand, a value of HR ≤ 1 indicates that there are no risks associated with the consumption of the minerals evaluated [16].

### 2.3. Animals

We used 24 8-week-old male BALB/c mice, free of pathogens weighing 20–25 g. The animals were kept in micro ventilated cages, at a controlled environmental temperature of 22 ± 1 °C, with a relative humidity of 40–60% and a photoperiod of 12 h dark/12 h light. The present work was approved by the Internal Committee of Animal Welfare in Teaching and Research (Comité Interno de Bienestar Animal en la Enseñanza e Investigación, CIBAEI) and the Biosafety and Hygiene committee (Comité de Bioseguridad e Higiene, CBSH) of the School of Veterinary Medicine and Zootechnics of the Autonomous University of Nuevo León, in addition to complying with the guidelines established in the Mexican norm NOM-062-ZOO-1999 (SENASICA).

### 2.4. Experimental Design

The 24 mice were randomly separated into two groups (*n* = 12). The animals of both groups received the standard diet “Rodent lab chow 5001” from Purina, the first group received purified water (Control group, H_2_O), while the second group received BB, both treatments were administered ad libitum for 10 days.

After 10 days of treatment, 6 mice from each group (H_2_O pre-UC and BB pre-UC) were sacrificed in order to evaluate the effect of BB in healthy animals. The other 6 animals of each group (H_2_O post-UC and BB post-UC) underwent experimental induction of UC by the intrarectal administration of 2 mg of 2, 4, 6-trinitrobenzene sulfonic acid (TNBS), dissolved in 4% acetic acid (AcOH), in a final volume of 0.150 mL [21].

The animals were fasting for 24 h before the induction of UC [22,23,24]. Subsequently, the animals were anesthetized with ketamine/xylazine at a dose of 10–100 mg/kg, which was applied intraperitoneally [25]. The intracolonic injection was performed carefully through a plastic catheter with a diameter of 1 mm, the catheter was inserted 3 cm into the mouse colon for the administration of 0.150 mL of the solution of TNBS and AcOH, before taking the catheter out, 0.150 mL of air was applied in order to spread the solution completely in the colon [21]. Finally, the animals were held vertically with their heads down for 5 min, to ensure the distribution of the solution and avoid the return of this [26]. The animals were sacrificed 24 h after the induction of UC.

### 2.5. Sample Collection and Processing

Before sacrifice, the animals were anesthetized intraperitoneally with ketamine/xylazine at a dose of 10–100 mg/kg. Mice were euthanized by cervical dislocation [25,27,28].

For sample collection, the abdominal cavity was opened through a bilateral subcostal incision. The abdominal cavity was explored, to detect abnormalities. With the aid of a scalped, the entire colon was removed and washed with isotonic saline solution (NaCl 0.9%). The colon was divided into two sections by longitudinal cut. One of the sections was preserved at room temperature in a 4% paraformaldehyde (PFA) for histological analysis, and the other in RNAzol (Molecular Research Center, Inc., Cincinnati, OH, USA) to perform total RNA extraction.

### 2.6. Histological Analysis

The colon samples were embedded in paraffin blocks, which were cut in a microtome to obtain sections of 5 μm thick to perform Hematoxylin and Eosin (H&E) and Periodic Acid Schiff (PAS) strains.

Histological sections were observed under a light microscope (Nikon, Eclipse 50i, New York, NY, USA) at 40 X and 100 X to show and locate lesions, neutrophil infiltration, and morphological changes associated with the treatments. The results were recorded using the Q-capture Pro 7 program (QIMAGING ^®^, Tucson, Arizona, AZ, USA).

### 2.7. Cytokine Quantification by qPCR

Total RNA extraction was performed using RNAzol^®^ (Molecular Research Center, Inc., Cincinnati, OH, USA) following the manufacture´s instructions. The synthesis of complementary DNA (cDNA) was performed using the ImProm-II TM Reverse Transcription System (PROMEGA ^®^, Madison, WI, USA). The relative quantification of the sequences of interest was analyzed by quantitative PCR (qPCR) (7500 Real-Time PCR System Applied Biosystem) using SYBR Green reagent. The quantification of gene expression by qPCR was calculated using the ΔΔCt method. Gpd-1 was used as a control gene to normalize the expression of the genes of interest—IL-1β, IL-4, IL-6, IL-10, TNF-α, and INF-γ (Table 1). The samples of the group H_2_O pre-CU were used as a calibrator.

The amplification reaction was performed using the GoTaq qPCR Master Mix Kit (PROMEGA^®^, Madison, WI, USA). The reaction was performed in a final volume of 20 μL for each sample: 10 μL of Mix GoTaq qPCR (2×), 1 μL of the pair of primers of interest which were at a concentration of 100 μM, 75 ng of the cDNA of interest (1 μL), and finally, the reaction was completed with 8 μL of nuclease-free water. The qPCR reactions were performed in a 96-well PCR^®^ Microplate (Axygen Scientific^®^, Union City, CA, USA), covered with Platemax^®^ UltraClear Sealing Film (Axygen Scientific^®^, Union City, CA, USA). Each sample was placed in triplicate on the plate; moreover, we put a control reaction which consisted of adding all the reagents used to make the amplification reaction except the cDNA.

The synthesis conditions used were as follows: an initial cycle at 50 °C for 2 min, one cycle of polymerase activation at 95 °C for 10 min, followed by 40 cycles with a denaturation temperature at 95 °C for 15 s, an alignment of 60 °C by 60 s and an extension at 72 °C by 45 s. At the end of the synthesis program, the temperature was increased from 60 to 95 °C to create a dissociation curve.

### 2.8. Statistic Analysis

Statistical analyzes were performed using IBM SPSS Statistics 22 (IBD Corporation, New Orchard Road Armonk, New York, USA). Results are presented as the mean ± standard error. Gene expression data were analyzed with a one-way analysis of variance (ANOVA) followed by Tukey’s multiple comparison test of means. Values of *p* < 0.05 were considered statistically significant. The GraphPad Prism 6 program was used to generate graphics.

## 3. Results

### 3.1. Nutritional Content of the Bone Broth

From every 100 mL of BB was obtained 333 mg of solid sample (Table 2). The results show that 7.5% of the solids present in the BB correspond to minerals and 74.62% to protein. Because the importance of the protein consumed in the diet is due to its ability to provide AA, it was determined that 69.92% of the solids of the BB are AA.

#### 3.1.1. Nutritional Contribution of Amino Acids from Bone Broth

A portion of 100 mL of BB provides 232.8671 mg of AA, 54.56% corresponds to essential amino acids (EAA) and 44.43% to non-essential amino acids (NEAA) (Table 3). The main AA found in BB were Glu, His, Arg, Asp, Lys, Gly, Thr, and Val, which are at a concentration higher than 14 mg/100 mL of BB, those of lower concentration were Ala and Ile. The most abundant EAA found in BB were His, Arg, and Lys (43.98, 17.28, and 15.08 mg/100 mL of BB, respectively). The most abundant NEAA found in BB were Glu, Asp, and Gly (50.14, 16.74, and 15.01 mg/100 mL of BB, respectively).

#### 3.1.2. Nutritional Contribution of Minerals from Bone Broth

The mineral content of BB obtained from bovine femur is presented in Table 4. The main minerals found in the BB were Na, Ca, P, K, and Mg; these were found in amounts greater than 1 mg for every 100 mL of BB. On the other hand, the minerals found in lower concentrations were Co and Mn (0.0022 y 0.0004 mg/100 mL of BB). In addition, the analysis was performed for Mo, which could not be detected because the technique is capable of detecting minerals in a minimum concentration of 0.0001 mg/100 mL, which indicates that this mineral is found in lower amounts.

Because the bones contain some minerals that can be toxic to the body depending on the dose at which they are found, the HR was calculated based on the mg of minerals present in 100 mL of BB, and the recommended daily intake; we obtained HR values of ≤1, which indicates that there are no risks associated with the consumption of any of the minerals we evaluated (Table 4).

### 3.2. The Bone Broth Prevents Histological Damage in the Murine Model of Ulcerative Colitis

Using H&E staining, it was determined that colon in the animals of the H_2_O pre-UC group showed a normal architecture: intact epithelium, preserved mucosa and submucosa, and well-defined Lieberkühn crypts (Figure 1A). Furthermore, the samples were analyzed at 100× to evaluate the absence of inflammatory cells and signs of hemorrhage (Figure 1B). These results were considered as a negative control to determine that the reagents and the dose used to establish the UC model were effective. Figure 1C,D show the results obtained from the animals of the BB pre-UC group, we observed similar results to the negative control, these results indicate that BB consumption does not generate morphological damage in the colon.

Figure 1E,F shows the results obtained from the animals of the H_2_O post-UC group. We observed a severe architectural distortion of the colon, epithelial erosion, crypt distortion, and the presence of abundant neutrophils and erythrocytes. These results demonstrate the effectiveness of the protocol used to establish the model of UC; these results were taken as a positive control. Figure 1G,H shows that the animals of the BB post-UC group present a conserved epithelium, moderate distortion of the crypts, and moderate presence of erythrocytes and neutrophils.

Using the PAS histochemical staining, we determined the abundance of goblet cells, which show a purple/magenta coloration, and integrity of the epithelium at the apical edge of the enterocytes where the microvilli are found.

The groups H_2_O pre-UC (Figure 2A,B) and BB pre-UC (Figure 2C,D) showed a normal distribution of goblet cells and positive staining at the apical edge of the enterocytes.

The intrarectal administration of TNBS and AcOH reduced the number of goblet cells and affected the integrity of the colon epithelium; these results are shown in the H_2_O post-UC group (Figure 2E,F). On the other hand, the animals of the BB post-UC group showed a moderate loss of goblet cells, and some sections of the epithelium had slight staining in the apical edge of the enterocytes (Figure 2G,H).

### 3.3. The Bone Broth Modulates the Expression of Pro-Inflammatory Cytokines in the Murine Model of Ulcerative Colitis

Expression levels of pro-inflammatory cytokines increased significantly in the animals of the H_2_O post-UC group: IL-1β, IL-6 (*p* < 0.0001), and TNF-α (*p* < 0.05), compared to the BB post-UC, H_2_O pre-UC, and BB pre-UC. The expression level of INF-γ (*p* < 0.0001) had a significant increase in the groups that underwent UC induction (H_2_O post-UC and BB post-UC) compared to the groups, however, the animals that received the prophylactic treatment of BB (BB post-UC) (*p* < 0.05) presented the highest expression levels (Figure 3).

We analyzed the expression levels of the anti-inflammatory cytokines: *IL-4* (*p* < 0.0002) and *IL-10* (*p* < 0.0006), to evaluate whether BB has an anti-inflammatory effect mediated by the expression of these cytokines. A significant increase in the expression level of both cytokines was observed in the animals of the BB post-UC group, compared to the H_2_O post-UC, BB pre-UC, and H_2_O pre-UC group (Figure 4).

The prophylactic administration of BB in the murine model of UC (BB post-UC) reduced histological damage and prevented the expression of IL-1β by 61.12%, IL-6 by 94.70%, and TNF-α by 68.88%, increased the expression of INF-γ by 177.06%, IL-4 by 541.36% and IL-10 by 531.97%, concerning the control group that underwent UC induction (H_2_O post-UC) (Table 5).

## 4. Discussion

Bone broth is a rich source of AA, minerals, and proteins, for this reason, it has been used for years in the prevention and treatment of several diseases [1,15,17]. However, no study has focused on analyzing this effect in the prevention and treatment of diseases that can be caused by nutritional deficiencies, such as UC [3,4,5]. Currently, the drugs used for the treatment of this disease are based on the administration of anti-inflammatory drugs and immunosuppressants for long periods, the disadvantage is that they are expensive and have a low degree of effectiveness, accompanied by side effects [12,13]. Therefore, the purpose of the present study was to identify the nutrients present in BB and to determine if its consumption prevents clinical symptoms in a murine model of UC.

In the present work, femoral bones were used to perform the preparation of the BB, because products from animal origin such as meat, derived products such as bones and bone marrow are the main source of nutrients in the diet [14,29]. In addition, that the bone marrow provides more nutrients than the meat [30]. This is related to reports where it is estimated that the broth obtained from beef contributes 5.9789 mg of AA per 100 mL [31], while in the present study, it was found that 100 mL of BB contained 232.8671 mg of AA.

The characterization of the nutrients present in BB prepared from bovine femur revealed that BB is a food with a high nutritional value, 54.56% of AA corresponds to EAA; these results can be attributed to the type of bone used, in this case, the femur contains a large proportion of bone marrow, which is rich in EAA [29]. It can be attributed that the anti-inflammatory effect of BB is due to the EAA that it provides [32].

Certain AA are considered biomarkers for the diagnostic and treatment of UC patients, such as Gln and His, which are significantly lower in UC patients than in healthy people [33]. The present study revealed that 21.53% of the AA present in the BB correspond to Glu, while 18.89% to His. We can infer that consumption of BB could help increase and maintain the levels of Glu and His, by determining a dose of BB that can meet the necessary requirements of these AA.

In addition, Glu has been reported that can significantly increase cell proliferation, decreases apoptosis and inflammation in the mucosa of the colon, coupled with a reduction in the histopathological evidence of lesions characteristics of UC [34], and His can reduce the expression levels of *TNF-α* and *IL-6* [35]. Which agrees with the results obtained, the prophylactic administration of BB decreased the histological damage caused by intrarectal inoculation of TNBS and AcOH, being able to infer that the results obtained are attributable to the consumption of these AA, since these correspond to 40.42% of the total AA present in the BB.

Among the nutrients found in BB are minerals, some studies have shown that Ca^++^ and Mg^++^ levels are associated with cooking time, however, it has not been studied whether the long cooking time established for making BB represents a risk for increased levels of minerals that can cause side effects if it is consumed in large quantities [16]. In the present study, it can be ruled out that the consumption of BB may represent a health risk. The recommended daily mineral intake is 8241 mg [1,16], 100 mL of BB contributes 0.30% of the amount of minerals that must be daily consumed. Although 50.3% of the minerals found in BB correspond to Na^+^, this does not represent a health risk, 100 mL of BB contributes 0.62% of the recommended daily intake of 2000 mg [16].

Inflammation and morphological damage of the colon in UC have been identified to be caused mainly by increased expression of *IL-1β*, *IL-6*, *TNF-α,* and *INF-γ* [9]. The induction of CU by the intrarectal administration of TNBS and AcOH produces a significant increase in the levels of expression of these cytokines, whereas the prophylactic administration of BB prior to the induction of BB resulted in a decrease in the expression of *IL-1β, IL-6,* and *TNF-α*. This indicates that the anti-inflammatory effect of BB is attributable to its ability to reduce the expression of cytokines involved in the development of UC.

One of the most relevant results of the present work is that BB increased the expression of *INF-γ*, although it is considered a pro-inflammatory cytokine, it is also capable of increasing the microbicidal activity of macrophages [36], which could explain the increase in the expression of *INF-y*. We can infer that BB stimulates its expression in order to increase the immune and protective response against foreign agents. On the other hand, in some diseases, it has been identified that some Th1-type cells produce *INF-y* and are capable of producing *IL-10* [37], the expression of these cytokines could have a cooperative effect in reducing the expression levels of *TNF-α* [38]. Although currently the mechanisms that lead to the joint production of these cytokines have not been elucidated.

Deregulation of the immune response is another factor associated with the severity and progression of the disease [8], and inhibition of anti-inflammatory mechanisms has been observed, such as the secretion of the cytokines *IL-4* and *IL-10* [10]. The results obtained suggest that BB can promote the expression of these cytokines. Previous studies have reported that therapies that promote increased levels of *IL*-10 can decrease the damage seen in UC, even reversing this damage completely [39]. In contrast, we observed that the role of *IL-4* is controversial, due to the pleiotropic effect that this interleukin [40]. However, gene therapy based on *IL-4* and *IL-10,* or a combination of these, reduces the damage caused by intrarectal administration of TNBS [10]. We can infer that the therapeutic effect of BB is due to its immunomodulatory capacity by increasing the expression of anti-inflammatory cytokines and reducing the expression of pro-inflammatory cytokines.

Not knowing the mechanisms that lead to the development of the disease makes it difficult to use effective treatments to combat it. Currently, there are no diagnostic markers that support 100% the effectiveness of the treatments, including those proposed in this work, the data obtained at the moment allow us to infer that the therapeutic effect of BB is due to its capacity immunomodulatory by increasing the expression of anti-inflammatory cytokines and reducing the expression of pro-inflammatory cytokines. Later studies will determine the limits of consumption of this food that generates a complete recovery of the colon without generating side effects.

## 5. Conclusions

BB is a rich food in AA, mainly AEE with anti-inflammatory properties. BB reduces the histological damage caused by intrarectal administration of TNBS and AcOH in BALB/c mice because it is capable of modulating the immune response by decreasing the expression of pro-inflammatory cytokines, and stimulating the expression of anti-inflammatory cytokines. The results obtained so far represent scientific evidence of the therapeutic effect of BB against UC, subsequent studies will allow defining whether the consumption of BB could be a natural and low-cost alternative with results as effective as those obtained through gene and pharmacological therapy.

## Figures and Tables

**Figure 1 medicina-57-01138-f001:**
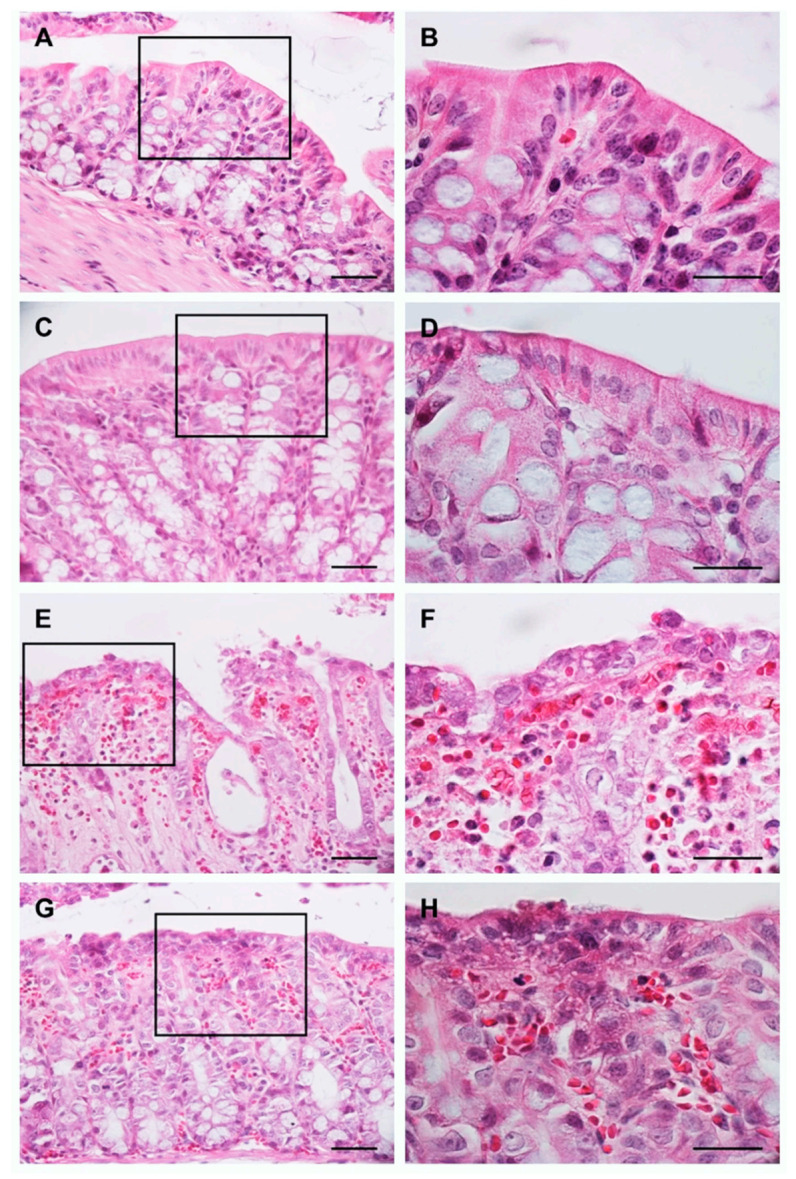
Evaluation of the morphology of the colon using the H&E stain. Images: (**A**,**C**,**E**,**G**) are at 40× magnification, images: (**B**,**D**,**F**,**H**) represent 100× magnification of the region marked with the rectangle in 40× images. Scale bar: 100 μm in all images.

**Figure 2 medicina-57-01138-f002:**
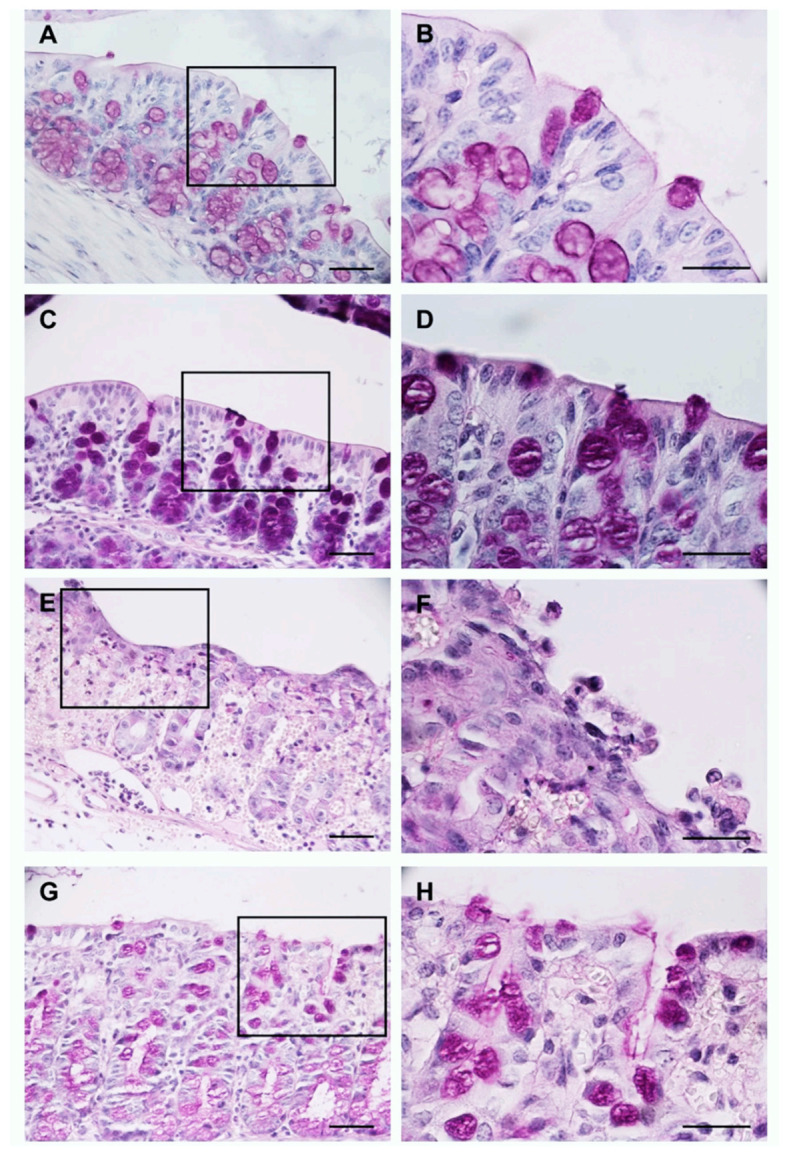
Evaluation of the morphology of the colon using the PAS histochemical stain. Images: (**A**,**C**,**E**,**G**) are at 40× magnification, images: (**B**,**D**,**F**,**H**) represent 100× magnification of the region marked with the rectangle in 40× images. Scale bar: 100 μm in all images.

**Figure 3 medicina-57-01138-f003:**
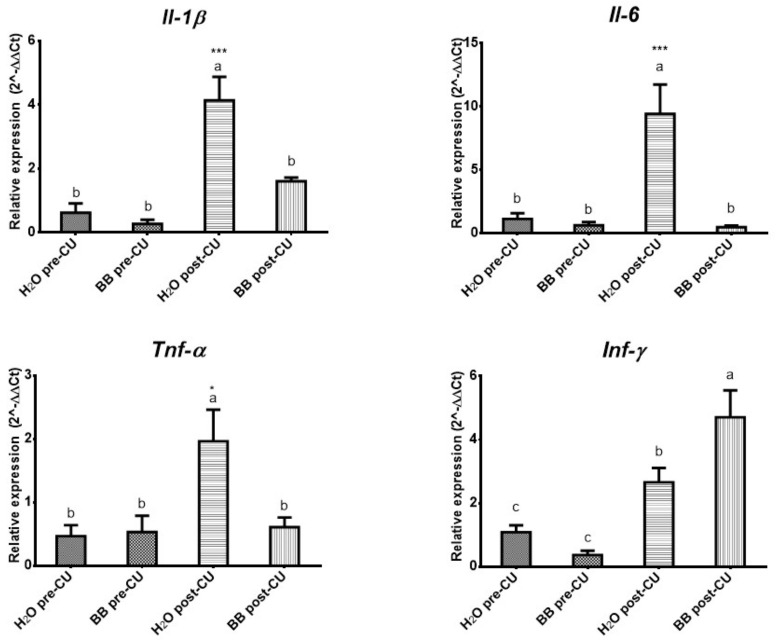
Evaluation of the expression of pro-inflammatory cytokines (2^-ΔΔCt). The data is presented as the mean of each group, the bars indicate the standard error, the letters on each column indicate the statistical differences. * *p* ≤ 0.05, *** *p* ≤ 0.001.

**Figure 4 medicina-57-01138-f004:**
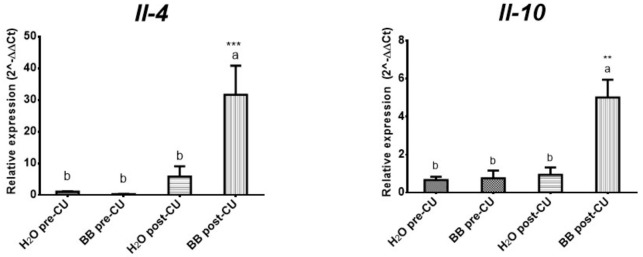
Evaluation of the treatments used on the expression of anti-inflammatory cytokines (2^-ΔΔCt). The data is presented as the mean of each group, the bars indicate the standard error and the letters on each column indicate the statistical differences. ** *p* ≤ 0.05, *** *p* ≤ 0.001.

**Table 1 medicina-57-01138-t001:** Primer sequences used for qPCR amplification of the genes of interest and the accession numbers of the sequences used for their design.

Gen	Genebank ID	Primers F (Forward), R (Reverse)
** *IL-1β* **	NM_008361.4	F: 5’ GGT ACA TCA GCA CCT CAC AA-3’ R: 5’ TTA GAA ACA GTC CAG CCC ATAC-3’
** *IL-4* **	NM_021283.2	F: 5’ TTG AGA GAG ATC ATC GGC ATT T-3’ R: 5’ CTC ACT CTC TGT GGT GTT CTT C-3’
** *IL-6* **	NM_031168.2	F: 5’ CTT CCA TCC AGT TGC CTT CT-3’ R: 5′ CTC CGA CTT GTG AAG TGG TAT AG-3′
** *IL-10* **	NM_010548.2	F: 5’ TTG AAT TCC CTG GGT GAG AAG-3’ R: 5’ TCC ACT GCC TTG CTC TTA TTT-3’
** *INF-γ* **	NM_008337.4	F: 5’ CTC TTC CTC ATG GCT GTT TCT-3’ R: 5’ TTC TTC CAC ATC TAT GCC ACT T-3’
** *GPD1* **	NM_010271.3	F: 5’ CCT ACT GCT GAC CTT TCT TCT C-3’ R: 5’ GCC CTG AGG ACG ATA AAC TAT AA-3’
** *TNF-a* **	NM_013693.3	F: 5’ TTG TCT ACT CCC AGG TTC TCT-3’ R: 5’ GAG GTT GAC TTT CTC CTG GTA TG-3’

**Table 2 medicina-57-01138-t002:** Nutritional content of bone broth obtained from bovine femur.

Nutritional Content	mg/100 mL
Protein	248.5181
AA	232.8671
Minerals	25.0176

**Table 3 medicina-57-01138-t003:** Amino acid content present in bone broth obtained from bovine femur.

AA	mg/100 mL
Asp	16.7499
Glu	50.1499
Ser	6.8265
Gly	15.0183
Ala	2.5974
Pro	3.3966
Cys	3.4632
Tyr	5.2614
Lys	15.0849
His	43.9893
Thr	14.7519
Arg	17.2827
Val	14.1858
Met	9.2907
Ile	3.0969
Leu	4.6620
Phe	7.0596
Total AA	232.8671
EAA	129.4039
NEAA	103.4632

**Table 4 medicina-57-01138-t004:** Calculation of the hazard ratio.

Minerals	mg/100 mL	Recommended Daily Intake (mg)	HR
Ca	6.4160	1000	0.0064
Mg	1.8460	400	0.0046
P	2.0370	1000	0.0020
Na	12.5840	2000	0.0063
K	1.9610	3800	0.0005
Fe	0.0430	18	0.0024
Cu	0.0310	1.7	0.0182
Zn	0.0970	14	0.0069
Mn	0.0022	5.5	0.0004
Co	0.0004	1.8	0.0002
Total	25.0176	8241	0.0030

**Table 5 medicina-57-01138-t005:** Gene expression profile of cytokines of the evaluated groups. Data are presented as the mean of each cytokine.

Cytokines	H_2_O Pre-UC	BB Pre-UC	H_2_O Post-UC	BB Post-UC
IL-1β	0.62 ± 0.30	0.27 ± 0.13	4.14 ± 0.73	1.61 ± 0.12
IL-6	1.14 ± 0.45	0.64 ± 0.25	9.43 ± 2.32	0.50 ± 0.12
TNF-α	0.47 ± 0.17	0.53 ± 0.26	1.96 ± 0.50	0.61 ± 0.14
INF-γ	1.08 ± 0.22	0.37 ± 0.14	2.66 ± 0.45	4.71 ± 0.85
IL-4	1.04 ± 0.13	0.26 ± 0.05	5.85 ± 3.20	31.67 ± 9.15
IL-10	0.67 ± 0.17	0.76 ± 0.41	0.94 ± 0.39	5.01 ± 9.15

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
