# Peer review of "Analysis of the Anti-Inflammatory Capacity of Bone Broth in a Murine Model of Ulcerative Colitis"

_medicina, 2021, doi:10.3390/medicina57111138_

Round 1
Reviewer 1 Report
The study presented in this manuscript brings interesting data on the possible anti-inflammatory role of bone broth in ulcerative colitis, using a murine model.
The paper is well written, following all the steps to present the work, the results, and the conclusions. The abstract presents the main aspects of the manuscript. The Introduction presents clearly the background and the need for this study. The Material and methods are very precise presented. The presentation of the results is well-realized, using also Tables and Figures of histological analysis. The Discussion is well conducted and leads to Conclusions supported by the data presented on the nutritional characteristics and anti-inflammatory role of bone broth in ulcerative colitis.
There are just a few minor changes that could improve the paper (mostly editing):
- use capital letters for IL, TNF, INF all over the manuscript
- line 302, ww not needed in wwdeficiencies
- verify sentence in line 328
Author Response
Enclosed you will find the respective answers to every point required by the referees regarding to the study: “Analysis of the Anti-Inflammatory Capacity of Bone Broth in a Murine Model of Ulcerative Colitis” with reference number medicina-1360933
All authors have read, reviewed and approved the corrected manuscript, also authors declare that they have no conflict interest nor competing interests.
Thanks in advance for your kind attention
Yours sincerely,
Uziel Castillo Velázquez. PhD
Laboratory of Veterinary Immunology
School of Veterinary Medicine and Zootechnics
Autonomous University of Nuevo León
General Escobedo, Nuevo León 66050, Mexico
Point 1.- “use capital letters for IL, TNF, INF all over the manuscript line 302” We appreciate the suggestion by the reviewer, considering the modification in the entire text, capitalizing each of the interleukins mentioned in the text.
Point 2.- “ww not needed in wwdeficiencies verify sentence in line 328” We appreciate the suggestion by the reviewer, making the modification in the text

Reviewer 2 Report
This is a very elegant, innovative study nicely presented.
However I would like the authors to comment on he limitations of the study.
I would also suggest to consider the conclusion of the consumption of bone broth can decrease the symptoms of UC, This is preliminary study based on animal model. It my opinion it is far to early to draw such a conclusion.
Author Response
Enclosed you will find the respective answers to every point required by the referees regarding to the study: “Analysis of the Anti-Inflammatory Capacity of Bone Broth in a Murine Model of Ulcerative Colitis” with reference number medicina-1360933
All authors have read, reviewed and approved the corrected manuscript, also authors declare that they have no conflict interest nor competing interests.
Thanks in advance for your kind attention
Yours sincerely,
Uziel Castillo Velázquez. PhD
Laboratory of Veterinary Immunology
School of Veterinary Medicine and Zootechnics
Autonomous University of Nuevo León
General Escobedo, Nuevo León 66050, Mexico
Point 1.- “ However I would like the authors to comment on he limitations of the study”.
We appreciate the suggestion by the reviewer, making the modification in the text and adding the following sentence (in red).
“Not knowing the mechanisms that lead to the development of the disease makes it difficult to use effective treatments to combat it. Currently, there are no diagnostic markers that support 100% the effectiveness of the treatments, including those proposed in this work, the data obtained at the moment allow us to infer that the therapeutic effect of BB is due to its capacity immunomodulatory by increasing the expression of anti-inflammatory cytokines and reducing the expression of pro-inflammatory cytokines. Later studies will determine the limits of consumption of this food that generates a complete recovery of the colon without generating side effects.”
Point 2.- I would also suggest to consider the conclusion of the consumption of bone broth can decrease the symptoms of UC, This is preliminary study based on animal model. It my opinion it is far to early to draw such a conclusion.
The reviewer is right the results are preliminary and not conclusive we modify the sentence “The results obtained suggest that BB can be used as an alternative therapy for the treatment of UC.” By “The results obtained so far represent scientific evidence of the therapeutic effect of BB against UC, subsequent studies will allow defining whether the consumption of BB could be a natural and low-cost alternative with results as effective as those obtained through gene and pharmacological therapy.”

This manuscript is a resubmission of an earlier submission. The following is a list of the peer review reports and author responses from that submission.